# "Doctor, when can I drive?" – Can we compensate an immobilization of the right wrist while driving a car: A pilot study

**Falk Hilsmann**[1], **Felix Lakomek**[1]*, **Max Prost**[1], **Dominique Schoeps**[1], **Ahmed Al Asadi**[2], **Erik Schiffner**[1], **David Latz**[1]

**1** Department of Orthopedic and Trauma Surgery, Medical Faculty and University Hospital Duesseldorf, Heinrich-Heine-University Duesseldorf, Duesseldorf, Germany, **2** Medical Faculty Duesseldorf, Heinrich-Heine-University Duesseldorf, Duesseldorf, Germany

* FelixNikolaus.Lakomek@med.uni-duesseldorf.de

## Abstract

### Introduction

The joints of the upper extremity are responsible for ensuring the safe movement of the body when steering and shifting gears. The impact of wrist immobilization and the subsequent movement limitations on driving ability remains inconclusively elucidated. The aim of the pilot study was to determine the range of motion required to safely operate a motor vehicle when the right wrist is immobilized. In addition, the compensators mechanisms that occur in this situation and enable safe driving to continue were to be Identified.

### Materials and methods

A total of 20 healthy subjects were studied as part of a driving simulation in a stationary driving simulator. The right wrist was immobilized, and all subjects were required to complete a standardized driving program consisting of representative driving maneuvers (A) shifting gear, B) left turns, C) right turns). To evaluate driving performance, speed, lane keeping, and shifting time were assessed using the driving simulator. In addition, the range of motion of the upper extremity, spine, and hip were collected using the motion capture system.

### Results

The average age of the 20 healthy participants was 28.2 years, and 40% were female. The elimination of the right wrist does not result in a significant reduction in driving performance. During the act of shifting gears, a significant decrease in extension was observed in the right elbow (p = 0.002; 95% CI [−6.13, −1.57]), while a significant increase in abduction in the right shoulder joint (p = 0.008; 95% CI [−7.46,

**Data availability statement:** Yes - all data are fully available without restriction; All relevant data can be found in the paper and the supplementary information.

**Funding:** ADAC Stiftung (Foundation) https://stiftung.adac.de. The funders had no role in study design, data collection and analysis, decision to publish, or preparation of the manuscript.

**Competing interests:** The authors have declared that no competing interests exist.

−1.28]) and flexion in the spine was observed (p = 0.011; 95% CI [−1.48, −0.22]). During a right-hand turn in the road, compensation occurs via the right elbow with a significant reduction in both maximum flexion (p = 0.008; 95% CI [2.98, 17.22]) and maximum supination (p = 0.005; 95% CI [−20.96, −4.34]). Conversely, when turning left, there is compensation via the left upper extremity: There is a significant increase in ulnar abduction of the left wrist (p = 0.03; 95% CI [−5.87, −0.33]) and minimal flexion of the left elbow (p = 0.012; 95% CI [1.22, 8.88]).

## Conclusion

Our data suggest that driving with an immobilized right wrist can be well compensated in healthy adults, although biomechanical changes in the upper extremity and spine were observed.

---

## Introduction

To operate a motor vehicle in a safe manner, it is necessary to possess a certain level of movement in addition to strength and coordination [1,2]. For a significant proportion of the population, the ability to operate a motor vehicle is fundamental to their sense of autonomy and self-determination [3,4] as well as their financial independence [5]. However, it has been determined that injuries and diseases of the musculoskeletal system can temporarily compromise a patient's ability to drive. Due to the variability of injury patterns, there remains a lack of consensus on the precise moment at which such injuries allow a return to driving. A representative patient survey revealed that patients primarily entrust the decision regarding their driving ability to their attending physician [6]. The present pilot study, therefore, seeks to address the question of whether a limitation of movement of the wrist could result in the revocation of driving ability.

The upper and lower extremities can be described as a kinematic chain, comprising the shoulder, elbow, and wrist joints or the hip, knee, and ankle joints [7,8]. The requisite range of motion (ROM) for safe driving has been previously investigated in studies conducted on healthy participants [2,9–12]. In particular, the upper extremity is crucial for executing intricate movement sequences, such as steering and shifting, which demand a high degree of precision. Nevertheless, the precise extent to which impaired mobility of the upper extremities compromises the ability to operate a motor vehicle in a safe manner remains to be conclusively determined. Prior research indicates that the limitation of one joint's range of motion can be partially compensated for by the capacity of adjacent joints [9,10,13]. To the best of the author's knowledge, only one study has attempted to define the necessary range of motion of the upper extremity for safe driving using a driving simulator and a motion-capture system [14]. A recent study focused only on the ability to compensate when driving with a restricted elbow [15]. Consequently, no study has hitherto analyzed the compensation mechanisms of the upper extremity as a functional unit (shoulder, elbow, wrist) while driving a car with a restricted right wrist using a driving simulator and

a motion-capture system. However, it can be postulated that with unrestricted mobility of the adjacent joints, a reduced range of motion, for instance at the wrist, can be effectively compensated for [16]. In contrast, a restriction of movement at the elbow is more likely to be tolerated poorly than a restriction of mobility at the adjacent joints [15,17]. Moreover, the contribution of the spine and the hip joint to the compensation of upper extremity movement restrictions during driving remains unclear.

The objective of this pilot study was to examine the compensation mechanisms and the alterations in the movement patterns of the adjacent joints of the upper extremity when driving a left-sided car with restricted movement of the right wrist. To determine the compensatory mechanisms of the upper extremity, movements in the spine and hip were also recorded. Additionally, the driving performance parameters were documented and analyzed under two distinct conditions (immobilization of the right wrist vs. free movement).

## Materials and methods

### Study design and recruitment

This pilot study was conducted in accordance with the ethical standards of the responsible committee on human experimentation (institutional and national) and was approved by the ethics committee prior to its initiation (ethical vote number 2021−1336). The study included 20 healthy participants with a valid driving license who drove their own car at least 5,000 km/year over the past three years. None of the participants suffered from acute or chronic injuries to the upper or lower extremities. In addition, the following exclusion criteria were selected: neurological disorders, active drug use, and lack of driving experience. All participants were adults, had been informed about the study, and had given written consent. The recruitment period commenced on May 1, 2021, and concluded on June 1, 2022.

To achieve the best possible comparability in basic research, the initial decision was made to include healthy participants. This excluded influencing factors such as reduced reaction time, chronic mobility restrictions, and visual impairment. Furthermore, only pain-free participants were selected, as it is difficult to assess the extent of impairment in each individual subject due to pain.

A power analysis was not performed prior to the study, as this is an exploratory study, and no comparable studies were available. Therefore, the sample size was based on similar basic research.

### Motion analysis and driving simulator

In this study, a motion-capturing system (Xsens, Netherlands; Rehagait Analyzer Pro, Germany; Hasomed GmbH) was utilized. The Xsens Motion Capture System from Movella is a sophisticated solution for precisely capturing human movement in real time. It uses inertial sensors that record movements without the need for external cameras or markers, making it particularly flexible and versatile. The driving analysis was conducted in a professional stationary driving simulator (Typ Trainer, FA Foerst, Germany). Following the provision of informed consent, the individual body anatomy of each participant was determined. Each participant was outfitted with an experimental suit equipped with motion tracking technology. The placement of the motion trackers was conducted in accordance with a standardized scheme based on anatomical landmarks. Each participant was required to undergo an initialization process to create an individual avatar. This experimental setup enables the simultaneous measurement of the upper and lower extremities and the spine. To evaluate the compensation mechanism of the upper extremity, only the shoulder, elbow, and wrist were taken into further analysis. To ascertain whether the compensation mechanism extends to the trunk, spine, and hip movements, an analysis was conducted. Two palm splints made of hard cast in different sizes without thumb inclusion were used for immobilization. A decision was made on an individual basis as to which of the two splints was used for each participant. Once the correct fit of all motion trackers had been verified, the study began.

## Procedure

To create a realistic yet standardized setting, participants were seated in a uniform position with a minimum of 25 cm between the body and the steering wheel. The distance between the head and the headrest was minimized, and the back was positioned as upright as possible [18]. To become acquainted with the operation of the driving simulator, each participant completed a standardized course with right-hand traffic for a period of three minutes.

To analyze the compensation mechanisms, each participant was required to complete the same standardized driving course on two occasions, once with a wrist orthesis (WO) and once without immobilization (no immobilization (NI)) of the right wrist. To prevent the habituation effect, the sequence was randomized using a random generator. The ride without immobilization was coded as 0, and the ride with the right wrist immobilized was coded as 1. The order was randomized individually for each volunteer. The following three scenarios were required to be completed on the course:

A) shifting

B) left turns

C) right turns

In Scenario A, the participants were required to shift into third gear and accelerate to a speed of 50 km/h. Subsequently, the vehicle was brought to a halt at the level of a road sign. Before scenario B, the transmission was switched to automatic. In scenarios B and C, the participants were then required to navigate a series of left and right turns, with the angles of these turns ranging from 90° to 180°.

## Collected data

Before the procedure began, informed consent was obtained from each participant, who was then required to complete a standardized questionnaire comprising personal data (age, gender, handedness, foothold, type of gearstick, and risk-taking, rated on a scale of 1–5), as well as the German version of the Arnett Inventory of Sensation Seeking (AISS-d).

The following parameters were identified as key determinants of driving performance: For maneuver A, the following parameters were determined: lane keeping (deviation to the left (-) and right (+) from the ideal line in meters (m)), speed (kilometers per hour (km/h)) and shifting time (time from start of driving to engaging third gear in seconds (s)). In the case of maneuvers B and C, only the variables of lane keeping, and speed were determined. In all cases, the mean value, the minimum value (min), and the maximum value (max) were determined; however, not all values were utilized in subsequent analyses. For instance, the minimum speed of 0 km/h was excluded from further analysis, as it did not contribute any insights that could be used to address the research question.

The motion-capturing system was employed to document the range of motion in the joints of the upper limb, spine, and hip across the three scenarios, with the objective of analyzing the movement and identifying potential compensation mechanisms. The range of motion was determined by considering the following degrees of freedom for the respective joints:

**Wrist:** ulnar/radial deviation (+/-), flexion/extension (+/-)

**Elbow:** flexion/extension (+/-), pro-/supination (>90°/<90°)

**Shoulder:** abduction/adduction (+/-), internal/external rotation (+/-), ante-/retroversion (+/-)

**Spine:** rotation (in segment Th12/L1) left/right (+/-), flexion/extension (in segment L5/S1) (+/)

**Hip:** flexion/extension (+/-)

## Data and statistical analysis

The data obtained from the motion-capturing system and the driving simulator were synchronized and subsequently processed using Excel® (Microsoft Corporation, Redmond, Washington, United States). The objective of our statistical analysis was to investigate the influence of wrist immobilization on range of motion and driving performance, while accounting for inter-subject variability. Given the hierarchical structure of the data set, with multiple observations per participant, a linear mixed model was deemed an appropriate statistical tool to account for the potential non-independence of observations within each participant. The model was specified with a fixed effect for immobilization of the right wrist and a random intercept. The model parameters were estimated using the restricted maximum likelihood method. Based on the estimated marginal means, additional pairwise post hoc comparisons were performed to investigate the differences between the two groups (wrist restriction/no wrist restriction, maneuver A-C). The analysis was conducted using IBM SPSS Statistics, version 29.

## Results

The mean age of the participants was 28.2 years (standard deviation (SD): ±3.1), with eight females and 12 males. Most participants were right-handed (17/20) and 75% reported driving a manual vehicle. The overall risk-taking score was estimated to be 4.45 (SD: ±0.8), with the average value of the AISS-D score being 48.5 (SD: ±4).

### Driving performance

The assessment of the parameters utilized to ascertain driving capability revealed no significant alteration because of the immobilization of the right wrist (Table 1).

Table 1 summarizes driving performance metrics—lane deviation (m; left = −, right = +), speed (km/h), and shifting time (s)—comparing conditions without (NI) and with wrist immobilization (WO). It includes p-values and 95% confidence intervals for the differences between both conditions.

**Table 1. Driving performance.**

| A (shifting) | | NI | | WO | | |
|---|---|---|---|---|---|---|
| Lane keeping | min | −0.82 | ± 0.11 | −1.00 | ± 0.12 | p=0.14 [−0.06, 0.41] |
| | mean | −0.49 | ± 0.10 | −0.63 | ± 0.09 | p=0.091 [−0.02, 0.30] |
| | max | 0.12 | ± 0.06 | 0.06 | ± 0.05 | p=0.081 [−0.01, 0.13] |
| Speed | mean | 37.05 | ± 1.11 | 37.55 | ± 1.16 | p=0.53 [−2.15, 1.15] |
| | max | 47.30 | ± 1.29 | 47.82 | ± 1.36 | p=0.66 [−2.93, 1.90] |
| Shifting time | mean | 9.94 | ± 0.46 | 9.81 | ± 0.47 | p=0.73 [−681.26, 948.40] |
| B (left turns) | | NI | | WO | | |
| Lane keeping | min | −3.26 | ± 0.25 | −3.06 | ± 0.18 | p=0.42 [−0.72, 0.31] |
| | mean | −1.73 | ± 0.15 | −1.64 | ± 0.15 | p=0.38 [−0.32, 0.13] |
| | max | 0.14 | ± 0.13 | 0.31 | ± 0.13 | p=0.16 [−0.40, 0.07] |
| Speed | mean | 30.31 | ± 0.83 | 31.07 | ± 0.73 | p=0.42 [−2.68, 1.17] |
| | max | 42.16 | ± 1.19 | 43.07 | ± 1.28 | p=0.59 [−4.36, 2.54] |
| C (right turns) | | NI | | WO | | |
| Lane keeping | min | −2.16 | ± 0.30 | −1.87 | ± 0.11 | p=0.35 [−0.93, 0.35] |
| | mean | −0.19 | ± 0.14 | −0.28 | ± 0.11 | p=0.39 [−0.12, 0.29] |
| | max | 1.26 | ± 0.16 | 0.97 | ± 0.16 | p=0.055 [−0.01, 0.58] |
| Speed | mean | 22.54 | ± 0.78 | 21.98 | ± 0.73 | p=0.42 [−0.87, 1.98] |
| | max | 37.45 | ± 1.66 | 35.82 | ± 1.12 | p=0.33 [−1.80, 5.06] |

The results of the study demonstrated that there was no statistically significant difference between the driving performance of the subjects when they were required to perform maneuver A with and without immobilization of the right wrist. This finding was consistent across all three performance metrics: lane keeping, speed, and shifting time. Regarding left and right turns (maneuvers B and C), the statistical analysis likewise indicated that there was no statistically significant change in the parameters of lane keeping and speed.

## Change in movement patterns

Table 2 illustrates the notable alterations in the kinematic patterns of the wrist, elbow, shoulder, spine, and hip resulting from the immobilization of the right wrist. These values were determined for all three maneuvers (A-C).

Table 2 presents the significant differences in joint range of motion between conditions without immobilization (NI) and with wrist orthosis (WO) across maneuvers A–C. Measurements include wrist, elbow, shoulder, spine, and hip movements. The table reports only significant changes, providing the 95% confidence intervals for these differences, with p-values for significant effects shown in bold.

**Maneuver A (shifting).** The statistical analysis revealed a significantly reduced extension of the right elbow (no restriction: −3.05° vs. wrist immobilization: 0.80°; p = 0.002). About the right shoulder, the statistical analysis revealed a significantly greater minimum and maximum abduction (no restriction: 10.60° to 29.70° vs. wrist immobilization: 14.55° to 34.60° (minimum p = 0.047, maximum p = 0.008), with a shift to a higher mean abduction (no restriction: 18.46° vs. wrist immobilization: 22.83°; p = 0.008) (Fig 1). Regarding the spine, the statistical analysis revealed a significantly higher flexion in the L5/S1 segment (no restriction: 20.80° vs. wrist immobilization: 21.65°; p = 0.011).

However, no significant difference was found for the left upper extremity or for the hips.

**Maneuver B (left turns).** Statistical analysis revealed a significant increase in ulnar deviation (no restriction: 27.75° vs. wrist immobilization: 30.85°; p = 0.030). Regarding the elbow, statistical analysis revealed a significantly lower minimum flexion for the left elbow (no restriction: 38.95° vs. wrist immobilization: 33.90°; p = 0.012).

No significant differences were observed in the range of motion of the right elbow, shoulders, spine, or hips.

**Maneuver C (right turns).** Statistical analysis revealed a significantly lower maximum flexion of the right elbow (no restriction: 78.85° vs. wrist immobilization: 68.75°; p = 0.008), as well as a decrease in supination (no restriction: 45.20° vs. wrist immobilization: 57.85°; p = 0.005).

No significant differences were observed in the range of motion of the left wrist, the left elbow, the shoulders, spine, or hips.

## Discussion

### Driving performance

Driving performance has been the focus of numerous studies and reviews [1,13,19–30]. Most of these studies and reviews have focused on the return to driving after surgical procedures and injuries to the lower extremities [1,26–30]. To evaluate this, several measurements have been established. These include total braking time, brake response time, and braking force [31–33]. In addition, a series of clinical tests provided which can be used as an alternative to these measurements in everyday clinical practice. These tests include the standing and stepping test [34,35]. The aforementioned tests for the lower limb relate primarily to the execution of the braking process, which represents a unidirectional movement. However, steering and shifting gears and the associated function of the upper limb is a multidirectional activity, which is why testing is significantly more complex. One potential test would be to ascertain whether the patient is able to turn the steering wheel free or at least 100° in both directions [14,36]. In summary, there are only a few validated tests that a treating physician can use to determine whether a patient can drive safely again.

To be able to analyze the significance of wrist mobility on fitness to drive, the safety-relevant driving data was first examined.

**Table 2. Change in movement patterns.**

| A (shifting) | | | | NI | | WO | | |
|---|---|---|---|---|---|---|---|---|
| **Elbow** | right | flexion (+)/ extension (-) | min | −3.05 | ± 1.30 | 0.80 | ± 1.22 | **p=0.002** [-6.13, -1.57] |
| | | | mean | 35.70 | ± 2.82 | 37.89 | ± 3.74 | p=0.51 [-8.96, 4.58] |
| | | | max | 70.30 | ± 2.38 | 72.25 | ± 2.48 | p=0.21 [-5.09, 1.19] |
| **Shoulder** | right | abduction (+)/ adduction (-) | min | 10.60 | ± 2.10 | 14.55 | ± 2.02 | **p=0.047** [-7.85, -0.05] |
| | | | mean | 18.46 | ± 1.79 | 22.83 | ± 1.39 | **p=0.008** [-7.46, -1.28] |
| | | | max | 29.70 | ± 1.60 | 34.60 | ± 1.57 | **p=0.008** [-8.38, -1.42] |
| **Spine** | L5/S1 | flexion (+)/ extension (-) | min | 17.55 | ± 0.81 | 17.90 | ± 0.67 | p=0.26 [-0.98, 0.28] |
| | | | mean | 18.70 | ± 0.78 | 19.18 | ± 0.63 | p=0.079 [-1.02, 0.06] |
| | | | max | 20.80 | ± 0.78 | 21.65 | ± 0.69 | **p=0.011** [-1.48, -0.22] |
| **B (left turns)** | | | | NI | | WO | | |
| **Wrist** | left | ulnar (+)/ radial (-) deviation | min | −22.60 | ± 3.03 | −19.75 | ± 3.06 | p=0.48 [-11.17, 5.47] |
| | | | mean | 9.44 | ± 2.09 | 12.26 | ± 2.41 | p=0.092 [-6.15, 0.50] |
| | | | max | 27.75 | ± 2.29 | 30.85 | ± 2.55 | **p=0.030** [-5.87, -0.33] |
| **Elbow** | left | flexion (+)/ extension (-) | min | 38.95 | ± 3.10 | 33.90 | ± 3.81 | **p=0.012** [1.22, 8.88] |
| | | | mean | 55.89 | ± 3.00 | 53.43 | ± 2.60 | p=0.10 [-0.53, 5.47] |
| | | | max | 75.70 | ± 3.73 | 72.05 | ± 3.31 | p=0.23 [-2.49, 9.79] |
| **C (right turns)** | | | | NI | | WO | | |
| **Elbow** | right | flexion (+)/ extension (-) | min | 35.65 | ± 2.85 | 37.65 | ± 3.03 | p=0.33 [-6.23, 2.23] |
| | | | mean | 53.29 | ± 2.61 | 53.38 | ± 2.36 | p=0.93 [-2.24, 2.04] |
| | | | max | 78.85 | ± 3.84 | 68.75 | ± 2.57 | **p=0.008** [2.98, 17.22] |
| | right | pro- (>90)/ supination (<90) | min | 45.20 | ± 3.50 | 57.85 | ± 4.66 | **p=0.005** [−20.96, −4.34] |
| | | | mean | 75.32 | ± 3.55 | 78.48 | ± 3.86 | p=0.13 [-7.40, 1.06] |
| | | | max | 110.45 | ± 4.16 | 103.15 | ± 2.75 | p=0.081 [-1.00, 15.60] |

In previous studies, driving performance was examined in terms of lap time, lane keeping, speed, collisions, and reaction to sudden hazards [20–24]. Therefore, the present pilot study defined performance in terms of shift time, lane keeping, and speed in a representative driving maneuver with gear changes and cornering.

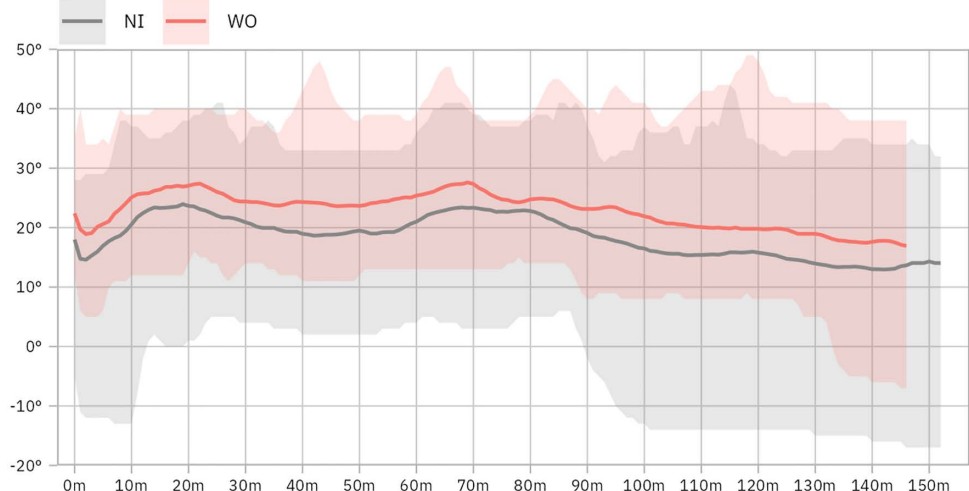

**Fig 1. Abduction and adduction of the right shoulder during shifting.** The graph shows right shoulder abduction (>0°) and adduction (<0°) during shifting. The grey area and line represent the range and mean motion without immobilization (NI), while the red area and line indicate motion with wrist orthosis (WO).

The effects of upper and lower arm casts on driving performance have been subject of numerous previous studies. A deterioration in driving performance was found in relation to immobilization using an upper arm cast [13,20,25]. The investigation of immobilization using a forearm cast splint did not show a significant decrease in driving performance, but several studies found a tendency towards poorer driving performance with forearm casts [13,19–21]. One study reported a significant reduction in driving performance, but this was limited to immobilization of the left side only [20].

It can be assumed that additional immobilization of the elbow leads to a decompensation of the kinematic chain. Gregory et al. assume that the performance level is maintained due to the increased alertness of the test subjects when driving with a plaster splint [21]. Schiffner et. all were the first to describe the changes in movement patterns of the upper limb when driving with a restricted right elbow. Restricted flexion or extension of the right elbow can be compensated for by changing the posture of the left arm and shoulders [15].

This finding contrasts with the subjective perception of the test subjects, who perceived a negative effect on driving performance even with a cast that only immobilized the wrist [23,24].

Our pilot study found no significant change in driving safety. There were no significant lane deviations or lane overshooting. A change in driving speed or the duration of gear changes was also not detected. This leads to the conclusion that immobilization of the right wrist has no influence on driving safety in the simulation scenario described in young healthy adults without pain. However, this may differ in patients who suffer from pain, reduced grip strength, or chronic movement restrictions, for example.

## Change in movement patterns due to immobilization

Since, as mentioned above, no limitation of driving safety was found in our study, the exact movement patterns were analyzed in the following to detect possible compensation mechanisms.

The upper limb can be considered as a kinematic chain including the wrist, elbow, and shoulder. If the right wrist is immobilized with a cast, it is imperative to ascertain the compensatory mechanisms that are responsible for compensating for the movement restriction. In most cases modifying the posture of the trunk and/or the movement on the ipsilateral

(right) or contralateral (left) upper extremity a compensation is reached. The adjustment of movement can be made at the proximal joint (in this case, the elbow) or at the most proximal joint (in this example, the shoulder joint). Alternatively, compensation can be achieved via the non-immobilized joint on the opposite side (i.e., the left wrist).

To understand which movements of the wrist, need to be compensated for during immobilization, it is first necessary to consider which directions of movement of the wrist are primarily required. A previous study demonstrated that dorsiflexion and ulnar abduction in the wrist are utilized significantly more frequently and beyond the active range of motion when operating a motor vehicle [9].

How these movements of the wrist, which are particularly responsible for the fine adjustment of the steering wheel position and the gearshift, were compensated for will be discussed in more detail below.

**Maneuver A (shifting).** In shifting gears in a straight line with the right wrist immobilized, we demonstrated that there is a notable alteration in movement in the adjacent joints on the same side and in the region of the spine. The compensation manifests as an increase in spinal flexion, an increase in the abduction of the right shoulder accompanied by a shift in the range of motion, and a reduction in elbow extension. No significant change was observed in the movement of the upper extremity contralateral to the immobilized wrist.

Our findings are in line with those of prior studies examining alterations in shoulder joint movement in patients with immobilized wrists and those who have undergone wrist arthrodesis [37–39]. The immobilization of the wrist resulted in a notable increase in shoulder abduction during activities such as feeding, stacking, and pouring [38,39]. Patients with a wrist arthrodesis also report significant difficulty with activities in confined spaces (e.g., changing a spark plug in a car) where compensation via the elbow or shoulder is not possible [37].

The observed increase in flexion in the spine was exclusive to the shifting gears maneuver, with no discernible change in the hip region. The shifting process represents a special movement sequence, a so-called bimanual activity. The hands perform different movements: The left hand stabilizes the steering wheel in the same position as without the cast, while the right hand operates the gearstick. When the wrist is immobilized with a cast, the spine and right shoulder bring the right hand into position, thereby performing the gross movements necessary for shifting. The right elbow also executes the smaller movements required to facilitate this action.

**Maneuver B (left turns) and C (right turns).** Depending on the direction of the turn, compensation is required when cornering with the right wrist immobilized. When cornering to the left, the left wrist and elbow are the primary joints utilized for compensation. A significant increase in ulnar deviation is observed in the left wrist, accompanied by a significant decrease in minimal flexion, namely a position of the left elbow in increased extension. This is because the majority of steering wheel turns to the left are executed with the left arm. To ensure safe driving on a left-hand turn, it is necessary to employ increased radial abduction in the left wrist and a more extended position of the left arm, given that the right hand is solely responsible for stabilizing the steering edge.

In the case of driving a right-hand turn, compensation is only possible via the more proximal joint, namely the right elbow. A significant reduction in the range of motion for both maximum flexion and supination was observed. Consequently, the position of the right wrist differs when steering to the right with the arm immobilized as opposed to driving without immobilization. It seems reasonable to posit that safe driving of the right turn was also possible without additional compensation via the left upper extremity, given that gripping the steering wheel was feasible despite the cast.

Our findings are in line with those of a prior study that observed no notable impact of a cast on steering capability [24]. However, an additional study revealed a notable elevation in the probability of deviating considerably from the prescribed route when a cast with a thumb enclosure was worn [23]. This can be attributed to the fact that steering with a cast, which lacks the ability to grip the steering wheel, is only possible due to friction or resistance from the cast and considerable force [13]. In our experimental setup, however, the plaster was without a thumb enclosure, and the steering wheel could be gripped with the plaster in place. The participants also did not suffer from pain and there was no reduction in grip strength. An additional potential explanation for the observed results when driving left and right turns is the possibility

of maintaining pro- and supination, as well as free movement of the shoulder joint, with the cast in place. It has been demonstrated that pro- and supination, in addition to shoulder flexion and rotation, are critical factors in the avoidance of potential hazards [14]. The act of pronation and supination results in the hands being positioned at the edge of the steering wheel. The role of the wrist in steering is relatively minor, with the shoulder joint making the largest contribution [14]. Existing movement restrictions could have an impact on the described movement sequences for avoiding potential hazards. It therefore stands to reason that such restrictions could also have an impact on the compensatory mechanisms when driving with the right wrist immobilized.

## Limitation

The pilot study was conducted with only young healthy test subjects without pain or chronic movement restrictions. It has been demonstrated that pain [1,28] and the use of painkillers have an impact on driving performance [40]. In addition, injuries such as a distal radius fracture result in diminished grip strength and mobility [41]. This simulator-study focused exclusively on shifting gears and driving right and left-hand turns, whereby the turns were only performed with an automatic transmission. Thus, only a limited number of scenarios were examined, whereas reversing and shifting gears while cornering was not. However, it is postulated that the side of the steering wheel and the type of gear shift have an impact on driving performance when immobilized with a cast [36,42]. In addition to the limitations already mentioned, the generalizability of the pilot study results is limited due to the small and homogeneous sample size. Despite randomization, the influence of learning effects remains unclear.

Furthermore, no adjustment for multiple comparisons was made in the analysis.

## Conclusion

Immobilizing the right wrist with a plaster splint did not result in any discernible alteration in driving performance in young healthy adults in controlled conditions. Shifting gears with the right immobilized wrist the compensation is provided by the ipsilateral elbow and shoulder and the position of the spine. When cornering, the intended direction is critical. In a right turn, compensation occurs through the right elbow. In a left turn, compensation occurs through the left wrist and left elbow.

The combination of right-hand turn and simultaneous gear shifting is critical and likely to result in a decrease in driving performance. Safe driving with the right immobilized wrist might be possible, but this requires unrestricted function of the adjacent joints (kinematic chain), the left upper extremity and spine as well as painlessness and full grip strength. Currently, the available results cannot yet be applied to patients.

The results of this study should be incorporated into everyday clinical practice. If a patient with immobilization of the right wrist inquiries about driving, the physician is advised to conduct an examination of the upper extremity and the spine.

Further studies are needed to understand the effect of immobilization of the left wrist on driving performance. In a next step, studies on patients with real wrist injuries could be conducted to analyze the additional effects of pain and injury-related relief on driving performance. Other scenarios such as reversing and shifting when cornering must also be investigated.

## Supporting information

**S1 File. Participant1.**
(XLSX)

**S2 File. Participant2.**
(XLSX)

**S3 File. Participant3.**
(XLSX)

**S4 File. Participant4.**
(XLSX)

**S5 File. Participant5.**
(XLSX)

**S6 File. Participant6.**
(XLSX)

**S7 File. Participant7.**
(XLSX)

**S8 File. Participant8.**
(XLSX)

**S9 File. Participant9.**
(XLSX)

**S10 File. Participant10.**
(XLSX)

**S11 File. Participant11.**
(XLSX)

**S12 File. Participant12.**
(XLSX)

**S13 File. Participant13.**
(XLSX)

**S14 File. Participant14.**
(XLSX)

**S15 File. Participant15.**
(XLSX)

**S16 File. Participant16.**
(XLSX)

**S17 File. Participant17.**
(XLSX)

**S18 File. Participant18.**
(XLSX)

**S19 File. Participant19.**
(XLSX)

**S20 File. Participant20.**
(XLSX)

**S21 File. Participants.**
(XLSX)

**S22 File. Statistical analysis.**
(XLSX)

## Acknowledgments

We would like to thank all volunteers for taking part in the study.

## Author contributions

**Conceptualization:** Felix Lakomek, Max Prost, Dominique Schoeps, Ahmed Al Asadi, Erik Schiffner, David Latz.

**Data curation:** Falk Hilsmann, Felix Lakomek, Ahmed Al Asadi.

**Formal analysis:** Falk Hilsmann, Felix Lakomek, Dominique Schoeps, Ahmed Al Asadi, Erik Schiffner, David Latz.

**Funding acquisition:** David Latz.

**Investigation:** Falk Hilsmann, Felix Lakomek, Ahmed Al Asadi.

**Methodology:** Falk Hilsmann, Felix Lakomek, Max Prost, David Latz.

**Project administration:** Dominique Schoeps, Erik Schiffner, David Latz.

**Resources:** Falk Hilsmann, Felix Lakomek, Max Prost.

**Software:** Falk Hilsmann, Felix Lakomek, Max Prost, Dominique Schoeps, Ahmed Al Asadi, David Latz.

**Supervision:** Falk Hilsmann, Erik Schiffner, David Latz.

**Validation:** Felix Lakomek, Max Prost, Dominique Schoeps.

**Visualization:** Falk Hilsmann, Felix Lakomek, Max Prost, Ahmed Al Asadi.

**Writing – original draft:** Falk Hilsmann.

**Writing – review & editing:** Felix Lakomek, Erik Schiffner, David Latz.

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
