## [Decision Letter · Decision Letter 0]

28 Aug 2025

PONE-D-25-35823"Doctor, when can I drive?" – Can we compensate an immobilization of the right wrist while driving a car?PLOS ONE?

Dear Dr. Hilsmann,

Thank you for submitting your manuscript to PLOS ONE. After careful consideration, we feel that it has merit but does not fully meet PLOS ONE’s publication criteria as it currently stands. Therefore, we invite you to submit a revised version of the manuscript that addresses the points raised during the review process.

**ACADEMIC EDITOR:**

We look forward to receiving your revised manuscript.

Kind regards,

Priti Chaudhary, M.S.

Academic Editor

PLOS ONE

Journal Requirements:

2. In the online submission form, you indicated that [The data cannot be made publicly available due to the data size of the raw data and the stepwise processing. The data, including the raw data, are available on request from the corresponding author for researchers who meet the criteria for access to confidential data.

Reviewers' comments:

Reviewer's Responses to Questions

**Comments to the Author**

1. Is the manuscript technically sound, and do the data support the conclusions?

Reviewer #1: Yes

Reviewer #2: Yes

Reviewer #3: Yes

2. Has the statistical analysis been performed appropriately and rigorously?

Reviewer #1: Yes

Reviewer #2: I Don't Know

Reviewer #3: N/A

3. Have the authors made all data underlying the findings in their manuscript fully available?

Reviewer #1: No

Reviewer #2: Yes

Reviewer #3: Yes

4. Is the manuscript presented in an intelligible fashion and written in standard English?

Reviewer #1: Yes

Reviewer #2: Yes

Reviewer #3: Yes

Reviewer #1: The introduction well structured with clear aim. Combining the aim with statistics related to road accidents would have been a plus

The methods please add a brief description of the motion capturing system used

The limitation and conclusion are an add to the paper

Reviewer #2: Thank you for giving me the opportunity to review this valuable work.

The topic is clinically meaningful, and the manuscript provides new insights into compensatory kinematics. Overall, the work is good and promising, but several sections need clearer reporting, correction of inconsistencies, and tempering of conclusions before publication can be considered.

In the abstract, please include the number of participants, mean age, and sex distribution in the results section.

The study objective sentence is too long and difficult to follow.

Sample size justification is missing, please provide a sample size calculation or state that the study is exploratory.

Randomization procedure unclear, Describe how conditions were randomized.

Limitations include the lack of adjustment for multiple comparisons, a small sample size, and concerns about simulator generalizability. Additionally, potential learning effects should be considered.

Reviewer #3: Overall Assessment

The manuscript addresses an interesting and clinically relevant question concerning the impact of wrist immobilization on driving ability. This is a pertinent topic in orthopedic and trauma care, as patients frequently inquire about their fitness to drive following immobilization or surgery. The use of a driving simulator combined with motion capture analysis is a clear strength and adds objectivity to the study. The paper is well-structured, the methodology is described in detail, and the results are generally clear. However, several methodological, analytical, and reporting issues require clarification and revision before the work can be considered for publication.

Strengths

1. Novelty and Relevance: The study focuses on an underexplored area—upper extremity immobilization and compensatory strategies during driving.

2. Methodological Rigor: Use of motion capture and a standardized driving simulator protocol provides objective kinematic data.

3. Clinical Importance: Results may aid physicians in counseling patients regarding return to driving after wrist immobilization.

4. Clear Structure: The manuscript follows a logical format with clear separation of Introduction, Methods, Results, and Discussion.

Major Concerns

1. Sample Size and Power

o Only 20 participants were included, all of them healthy volunteers without actual injury or pain. While the study demonstrates biomechanical compensation, the small and homogeneous sample raises questions about generalizability.

o A power analysis to justify the sample size is not provided.

2. External Validity

o The study used healthy young adults (mean age ~28 years) without pain or disability. This limits applicability to real patients who often have pain, swelling, decreased grip strength, or comorbidities.

o This limitation is acknowledged but not sufficiently emphasized in the Discussion. A stronger statement is needed about how this affects clinical translation.

3. Simulator Limitations

o Only three driving scenarios were tested (shifting, left turn, right turn), which may not represent the complexity of real-life driving (e.g., emergency braking, rapid steering corrections, highway driving).

o Automatic transmission was used in some parts, potentially underestimating the role of the immobilized wrist.

4. Statistical Analysis

o The choice of linear mixed models is appropriate, but details of the model specification are sparse.

o It is not clear if adjustments for multiple comparisons were performed, given the large number of joint angles and outcomes analyzed. This raises the risk of type I errors.

o Effect sizes and confidence intervals are not presented, which would strengthen the interpretation.

5. Clinical Implications

o The conclusion that “safe driving with the right immobilized wrist is possible” may be overstated given the limited scenarios and lack of patient population.

o A more cautious interpretation is needed, highlighting that these findings apply only to healthy individuals in controlled conditions.

Minor Concerns

1. Abstract

o The abstract is generally well written but should explicitly state that only healthy volunteers were studied.

o Reporting of key statistical values (p-values, confidence intervals) would strengthen clarity.

2. Introduction

o While background references are provided, the review of existing literature could be more concise. Some citations appear repetitive.

o Clarify how this study advances beyond earlier work on elbow immobilization and driving.

3. Methods

o More information about randomization (order of immobilized vs. non-immobilized trials) is needed.

o Describe the wrist immobilization device more clearly—was thumb immobilization included? How standardized was the splint?

o Provide explicit exclusion criteria (e.g., musculoskeletal disorders, neurological conditions).

4. Results

o Tables are informative but could be simplified; some parameters (e.g., minimum/maximum values) may not add substantial value.

o Figures would benefit from clearer legends and consistent labeling (NI = no immobilization; WO = with immobilization).

5. Discussion

o The discussion is comprehensive but occasionally repetitive.

o Greater emphasis on the limitations (young healthy participants, lack of pain, limited scenarios) is necessary.

o Some references are dated; ensure that the most recent literature (past 5 years) is adequately included.

6. Language and Style

o Overall, the manuscript is well written. Minor grammatical edits and simplification of long sentences would improve readability.

Suggestions for Improvement

• Include a power analysis or at least a rationale for the sample size.

• Report effect sizes and confidence intervals along with p-values.

• Clarify whether correction for multiple comparisons was applied.

• Strengthen the limitations section, particularly regarding generalizability to real patients.

• Revise the conclusion to be more cautious, stressing that results cannot be directly generalized to injured or post-operative populations.

• Improve clarity of tables and figures, and ensure consistent terminology.

Conclusion and Recommendation

This is a well-designed pilot study that provides valuable biomechanical insights into compensatory mechanisms during driving with wrist immobilization. However, methodological clarifications, more cautious interpretation, and improvements in reporting are needed.

**Do you want your identity to be public for this peer review?** For information about this choice, including consent withdrawal, please see our Privacy Policy

Reviewer #1: No

Reviewer #2: No

Reviewer #3: No

---

## [Author Response · Author response to Decision Letter 1]

24 Oct 2025

Dear academic editor and reviewers,

Thank you for reviewing our manuscript and for your valuable, constructive suggestions for improvement.

Academic Editor

We address your comments below:

1) We have revised the manuscript regarding the style requirements. If further adjustments are necessary, we will be happy to make them upon specific notification.

2) We would like to make the data underlying our study available to other researchers as supplementary information.

3) The ORCID ID of the corresponding author, Felix Lakomek, is: 0009-0000-1074-0246.

4) This point does not apply to our manuscript and was therefore not considered.

Reviewer #1:

We have added a description of the motion capturing system (Xsens).

Reviewer #2:

As requested, we simplified the sentence about the study's objective in the abstract and expanded the information in the results section.

We also addressed the sample size and, as you suggested, explained that this is an exploratory study. We described the procedure for randomizing in more detail and added your points about the study's limitations.

Reviewer #3:

First, we would like to address your most important concerns:

1. Sample size and power

We decided to include healthy volunteers to achieve the best possible comparability in basic research. This excluded influencing factors such as reduced responsiveness, chronic mobility restrictions, and visual impairment. Furthermore, we selected volunteers who were pain-free, as it is difficult to assess the extent to which pain affects each individual volunteer.

A power analysis was not performed prior to the study because it’s an exploratory study and no comparable studies were available.

Therefore, we based the sample size on the basic research studies conducted by our research group (Latz, D; Schiffner, E; Schneppendahl, J, et al.: Doctor, when can I drive? -Range of functional ankle motion during driving. Foot and ankle surgery: official journal of the European Society of Foot and Ankle Surgeons. 10.1016/j.fas.2019.12.006; Latz, D; Schiffner, E; Schneppendahl, J, et al.: Doctor, when can I drive? - the range of elbow motion while driving a car. J Shoulder Elbow Surg. 28: 1139-1145. 10.1016/j.jse.2018.11.053; Latz, D; Schiffner, E; Schneppendahl, J, et al.: Doctor, when can I drive? - Range of motion of the knee while driving a car. The Knee. 26: 33-39.10.1016/j.knee.2018.11.005; Latz, D; Pfau, S; Koukos, C, et al.: Doctor, when can I drive. Obere Extremität. 12: 234-241. 10.1007/s11678-017-0426-0).

2. External Validity

You are right about the limited applicability to real patients. Therefore, we have adjusted the limitations and conclusion sections.

3. Limitations of the Simulator

The volunteers completed a scenario that required shifting gears and making left and right turns. These are typical maneuvers required when driving a car. Further investigation of the maneuvers you mentioned, such as rapid steering corrections, is of clinical relevance. However, the aim of this study was not to make a statement about reaction time and the lower extremities.

During cornering, only an automatic transmission was initially used in basic research. When to shift gears while cornering is an individual decision of the driver, which would lead to poor comparability. As you noted, this can lead to the role of the immobilized right wrist being underestimated. In the conclusion, we explicitly refer to the critical combination of shifting gears and simultaneous steering.

4. Statistical Analysis

No adjustment for multiple comparisons was performed. This is now mentioned under limitations.

The 95% confidence intervals for difference were added as requested. Reporting effect sizes is very useful from a statistical perspective, as you already mentioned.

Since our analysis was conducted with SPSS, and SPSS does not allow the calculation of effect sizes for the models used, the effect size could not be calculated.

5. Clinical Implications

The conclusion has been adjusted to include a more cautious interpretation of our results under controlled conditions in healthy individuals.

We now address your minor concerns:

1. Summary

The summary now explicitly states that only healthy volunteers were included. Furthermore, the 95% confidence intervals for difference have been added.

2. Introduction

This study differs from previous work on wrist immobilization by combining a driving simulator with a motion capture system. A comparable study on the elbow joint has already been published by our research group (Schiffner E, Lakomek F, Hilsmann F, Schoeps D, Prost M, Beyersdorf C, Windolf J, Latz D. Doctor, when can I drive? - compensation capability while driving with restricted elbow - a biomechanical analysis. JSES Int. 2024 Nov 1;9(2):542-548. doi: 10.1016/j.jseint.2024.09.028. PMID: 40182246; PMCID: PMC11962544). This was clarified again in the introduction.

3. Methods

The order of the respective rides with the right wrist immobilized and the right wrist not immobilized was randomized using a random generator. The ride without immobilization was coded as 0, and the ride with the right wrist immobilized was coded as 1. The order was randomized individually for each volunteer.

The right wrist was immobilized using a palmar hard cast splint without thumb inclusion.

The following exclusion criteria were chosen musculoskeletal disorders, neurological disorders, active drug use, and lack of driving experience.

4. Results

In our opinion, the tables have already been reduced to a minimum; further simplification would, in our opinion, result in a loss of information. For example, in Table 1, the parameter "minimum" was omitted for speed.

The legends and labels of the tables and figures have been adjusted.

5. Discussion

A stronger emphasis on the limitations has been made as requested.

A recent literature review was conducted to compile this discussion, and we believe the literature is comprehensive regarding the topic being discussed.

6. Language and Style

Changes have been made to improve readability.

We hope that the implementation of your constructive criticism has led to a further improvement in the quality of the manuscript. Thank you very much.

With kind regards,

Falk Hilsmann

---

## [Editor Report · Decision Letter 1]

11 Dec 2025

PONE-D-25-35823R1"Doctor, when can I drive?" – Can we compensate an immobilization of the right wrist while driving a car?PLOS One?

Dear Dr. Hilsmann,

Thank you for submitting your manuscript to PLOS ONE. After careful consideration, we feel that it has merit but does not fully meet PLOS ONE’s publication criteria as it currently stands. Therefore, we invite you to submit a revised version of the manuscript that addresses the points raised during the review process.

We look forward to receiving your revised manuscript.

Kind regards,

Priti Chaudhary, M.S.

Academic Editor

PLOS One

Journal Requirements:

**Additional Editor Comments:**

After revisions done by authors,The manuscript may be accepted as Pilot study; as the small sample size, the exclusive inclusion of healthy volunteers without pain or injury, and the limited range of simulated driving tasks reduces the generalizability to real patients.

---

## [Author Response · Author response to Decision Letter 2]

4 Jan 2026

Dear Sir or Madam,

Thank you for reviewing our revised manuscript. We are pleased to submit a revised version of our manuscript in accordance with your points of criticism, to meet the required publication criteria.

We will now briefly address the points of criticism you raised:

As mentioned in our previous letter, there was no specific recommendation regarding the citation of particular, previously published works. Therefore, no review for relevance and citability was conducted.

Furthermore, the bibliography was reviewed again for completeness and accuracy. No irregularities were found; in particular, no retracted works are cited.

Additional comments from the editors:

The manuscript has been revised as requested, including an adjustment of the title and other necessary changes.

We now hope that the manuscript will be accepted as a pilot study.

Thank you in advance.

Sincerely,

Falk Hilsmann

---

## [Editor Report · Decision Letter 2]

11 Jan 2026

"Doctor, when can I drive?" – Can we compensate an immobilization of the right wrist while driving a car: A pilot study

PONE-D-25-35823R2

Dear Dr. Falk Hilsmann,

We’re pleased to inform you that your manuscript has been judged scientifically suitable for publication and will be formally accepted for publication once it meets all outstanding technical requirements.

Kind regards,

Priti Chaudhary, M.S.

Academic Editor

PLOS One
---

## [Editor Report · Acceptance letter]

PONE-D-25-35823R2

PLOS One

Dear Dr. Hilsmann,

I'm pleased to inform you that your manuscript has been deemed suitable for publication in PLOS One. Congratulations! Your manuscript is now being handed over to our production team.

Kind regards,

on behalf of

Dr. Priti Chaudhary

Academic Editor

PLOS One